# Surgical Management of Primary Hyperparathyroidism—Clinicopathologic Study of 1019 Cases from a Single Institution

**DOI:** 10.3390/jcm9113540

**Published:** 2020-11-02

**Authors:** Jacek Gawrychowski, Grzegorz J. Kowalski, Grzegorz Buła, Adam Bednarczyk, Dominika Żądło, Zbigniew Niedzielski, Agata Gawrychowska, Henryk Koziołek

**Affiliations:** Department of General and Endocrine Surgery, Faculty of Medical Sciences in Zabrze, Medical University of Silesia, 40-055 Katowice, Poland; kowalskigrzeg@gmail.com (G.J.K.); gregor6007@onet.eu (G.B.); adambednarczyk90@gmail.com (A.B.); dominika.zadlo@gmail.com (D.Ż.); zbigniew.niedzielski@wp.pl (Z.N.); agatagawrychowska86@gmail.com (A.G.); henkoz@interia.pl (H.K.)

**Keywords:** primary hyperparathyroidism, parathyroidectomy, remedial surgery, ectopic mediastinal localization, persistent hypercalcemia

## Abstract

Background: Primary hyperparathyroidism (pHPT) is an endocrine disorder characterized by hypercalcemia and caused by the presence of disordered parathyroid glands. Parathyroidectomy is the only curative therapy for pHPT, but despite its high cure rate of 95–98%, there are still cases where hypercalcemia persists after this surgical procedure. The aim of this study was to present the results of a surgical treatment of patients due to primary hyperparathyroidism and failures related to the thoracic location of the affected glands. Methods: We present a retrospective analysis of 1019 patients who underwent parathyroidectomy in our department in the period 1983–2018. Results: Among the group of 1019 operated-on patients, treatment failed in 19 cases (1.9%). In 16 (84.2%) of them, the repeated operation was successful. In total, 1016 patients returned to normocalcemia. Conclusions: Our results confirm that parathyreoidectomy is the treatment of choice for patients with primary hyperparathyroidism. The ectopic position of the parathyroid gland in the mediastinum is associated with an increased risk of surgical failure. Most parathyroid lesions in the mediastinum can be safely removed from the cervical access.

## 1. Introduction

Primary hyperparathyroidism (pHPT) is a type of endocrine disorder resulting in increased secretion of the parathyroid hormone from abnormal parathyroid glands. It leads to serious disturbances in calcium and phosphate metabolism leading to elevated levels of calcium in blood serum which may be a risk factor for developing a hypercalcemic crisis [1,2]. Primary hyperparathyroidism is a relatively common disorder that is diagnosed in 0.1 to 1% of the general population, and more often in postmenopausal women [2]. In patients with pHPT, the parathormone (PTH) excess originates from neoplastic or hyperplastic cells of parathyroid parenchyma. Most cases involved a single abnormal parathyroid gland located in an usual neck site, but many of such changes occur in other sites including the thymus gland, the retroesophageal space or the thyroid gland. Sometimes, parathyroid lesions have been reported in the pericardium or soft and adipose tissues of the mediastinum up to the angle of the jaw [3,4]. The differentiation between hyperplasia and adenoma may not be easy even for an experienced surgeon. In case of hyperplasia, it is essential to the establish not only the size, but also the symmetry of affected glands. In many cases, even affected glands can appear normal in the gross appearance. This points to the importance of a biopsy taken from the tissue and its examination by a pathologist. Parathyroidectomy (PTX) is the procedure of choice for patients suffering from pHPT with a cure rate of 95–98%. Such an operation is now the procedure of choice at experienced surgical centers, with the use of an intraoperative PTH level (iPTH). However, even then the PTX for ectopic mediastinal lesions is a challenging procedure [4]. We reported a group of 1019 patients operated on for pHPT between the years 1983 and 2018.

## 2. Materials and Methods

This is a retrospective study of 1019 patients operated on between 1983 and 2018 (between 1983 and 2004 318 patients were operated on and from 2005 to 2018 701 patients were operated on) (Figure 1) for primary hyperparathyroidism (pHPT). The sample consisted of 759 women (74.5%) and 260 men (25.5%) between the ages of 19 and 81. We evaluated 953 operated-on patients during the same time period for parathyroid lesions located in the neck and carried out comparisons with a group of 66 patients with ectopic mediastinal lesions. Patients were identified based on operation protocols and histopathological findings regarding the number of lesions, their localization, and biochemical and surgical results. The diagnostic management consisted of a careful case history and physical examination, and routine biochemical examinations. The biochemical examinations included measurements of the serum calcium levels and intact PTH. Parathyroid localization studies included up until 2004 the ultrasound and subtraction technique (also known as dual-isotope imaging); from 2005 sestamibi technetium-99 m scintigraphy was used and from 2010, the single-photon emission computed tomography (SPECT) technique has usually been utilized, and rarely the CT scan. Two positive imaging studies were always required. Upon their discharge from the hospital, the patients were seen at least four times a year—i.e., every three months within the first two years and every six months thereafter. Physical and biochemical examinations were also performed. We arbitrarily divided cases of hypercalcemia after prior surgery into persistent (defined as hypercalcemia recurring within six months of the initial operation) or recurrent (hypercalcemia recurring after six months of normocalcemia).

## 3. Methods of Statistical Analysis

All of the collected data were included in a spreadsheet in Microsoft Office Excel 2019. The statistical analysis was performed in Statistica 12.5. After establishing and classifying the data using a Kolmogorow–Smirnow test, we used a Student’s *t*-test to compare data with a normal distribution and for the data non-normal distribution we used the Mann–Whitney U test. The results are presented as percentage changes, full numbers, means and standard deviations. The numerical data were compared with that of the Pearson chi-square test. The level of significance was calculated at *p* < 0.05. Effect size—ES—was determined using Youle’s Phi coefficient and Cramer’s V.

## 4. Results

Parathyroid lesions were identified in 1019 patients. Fatigue, muscle or bone weakness, and loss of appetite were the most commonly observed symptoms in 923 symptomatic patients with normal or ectopic localizations of dysfunctional parathyroid glands. No symptoms were recorded among the remaining 96 patients (Table 1).

The vast majority (93.6%) of parathyroid lesions were localized in the neck (predominantly within the left and right superior glands), whereas a minority (6.4%) were found in the mediastinum. We observed a significant difference in the occurrence of arterial hypertension among the patients with neck parathyroid glands when compared to mediastinal localization. Dermopathies were significantly more frequent in the group of patients operated on for mediastinal glands, but both effect sizes were weak.

Histological findings demonstrated benign lesions in 990 patients while malignancy was reported in the remaining 29 cases. Among the 1019 patients, we investigated 1226 lesions, with 856 of them related to a single gland and 163 impacting two or more glands (Table 2). Among multiple lesions, the affection of two glands was the most prominent, while affection of four glands was sporadic. The benign neoplasms have been further differentiated into hyperplasia—which were identified most often (827 lesions)—and adenomas, which made up the remaining 370 lesions.

The analysis of biochemical examination results indicated elevated serum calcium and parathyroid hormone levels within the entire group prior to operation without distinction of etiologies or localizations (Figure 2 and Figure 3).

Histopathological findings imply that hyperplasia and adenoma are the most common types of single and multiple lesions (Table 3). In the case of multiple lesions, the intercurrence of adenoma and hyperplasia of several glands occurred within the neck area, but it was not seen in mediastinal localization.

During performing operations, we discovered that the parathyroid gland most commonly affected within the entire group of patients was the inferior right, with 436 cases. The inferior left parathyroid gland was affected in 351 cases, superior left in 220 cases, and superior right in 219 cases. We also performed a comparison of the occurrence of neoplastic gland in patients with single-gland changes and those with multiple abnormal glands. In the case of patients with single-gland neoplasm, the most commonly affected was the inferior right parathyroid gland (347 cases) and subsequently the superior left (142 cases), the superior right (140 cases), and the inferior left (27 cases). Within the patients with more than one affected gland, abnormalities were found most often within the inferior right parathyroid gland (89 cases) and subsequently the superior right (79 cases), the superior left (78 cases), and rarely the inferior left parathyroid gland (24 cases). Double adenomas were situated most commonly in the lower right location (53%), subsequently the right (49%) and the upper left (30%) locations, and sporadically in the lower left location (17%), implying that double adenomas are usually found in the right part of the neck when parathyroid glands are not ectopic. The mean size of double adenomas was approximately 1.5 cm. After a parathyroidectomy was performed, no symptoms were prominent among almost all of the patients and both PTH and calcium levels decreased. Calcium levels has normalized postoperatively and remained within the normal range for 12 months or longer. 

A total of 19 patients required remedial surgery due to persistent or recurrent hypercalcemia. Out of the group of 1019 patients, failures in the surgical treatment of pHPT were reported in 19 cases—in eight cases, the lesions were located on the neck, and in 11 cases the lesions were in the mediastinum (Table 4).

Persistent hyperfunction was the cause of repeated operation in nine of them and relapse in 10. However, imaging studies failed to find the cause of the hyperfunction in three of them. The remaining 16 were operated on successfully (Table 5).

There were two (0.2%) cases of persistent hypercalcemia caused by lesions of the neck and some cases (10.6%) in the mediastinum. Recurrent hypercalcemia was found in four patients (0.4%) due to a neoplastic gland remaining in the neck area and three (4.5%) within the mediastinum. We did not find any changes in three cases, where two cases were related to a neck localization and one case to the mediastinum. The main cause of failure of the surgical treatment were parathyroid carcinomas (twice—the cause of persistent activity and four—recurrent).

## 5. Discussion

Parathyroidectomy is the treatment of choice in patients with primary hyperparathyroidism. The goal of PTX is to achieve normocalcemia by removal of the hyperfunctioning tissue that causes excessive secretion of parathyroid hormones. Surgical failures may result from an incomplete excision or parathyromatosis. Reoperative parathyroid surgery can be required in approximately 10% of patients [5]. Out of the 1019 patients, 19 (1.9%) received a surgery that failed. This mainly applies to patients with ectopic mediastinal lesions (16.6%).

We want to emphasize that 15% of our patients have a multiple gland disease and that the heterogenic group includes patients with multiple hyperplasia, double adenoma, hyperplasia as well as cancer. We believe that a visual indication of all parathyroid glands and excision or biopsy of the rest of them is an effective procedure for most patients operated on for pHPT. Moreover, in our opinion, an intraoperative measurement of PTH (iPTH) by a quick assay (QPTH) in predicting operative outcomes of parathyroidectomy is the procedure of choice. For the last 15 years we have measured QPTH in all operated-on patients. The goal of the surgical treatment is to remove hyperfunctioning glands. If hyperplasia is diagnosed, we remove three and a half parathyroid glands, or hyperfunctioning adenomas. Collar incision is a feasible approach to the mediastinal glands even at re-exploration. Among our 66 patients treated for a mediastinal ectopic gland, only four required thoracotomy. We must emphasize that about one-third of normal lower parathyroid glands are found below the lower thyroid pole, in the thyroid-thymic tract of fat or cervical tongue of the thymus. In our experience of reoperation for a “missing” gland, the thymus is its most probable location. Among seven of our patients reoperated on for persistent hypercalcemia, four glands were found within the thymus, one between the pericardial sac and the mediastinal pleura, close to the main left bronchus, one in the aorto-pulmonary window and one close to the right innominate vein and the superior vena cava. 

The analysis of surgical treatment failures for various parathyroid changes implies several explanations. These include the variable anatomy of parathyroid glands among different patients as well as ectopic locations of adenomas [6]. There are various opinions that all glands inferior to the superior border of the manubrium tend to be mediastinal [6,7]. Others consider that mediastinal parathyroid glands are those which are unreachable by cervical incisions [7,8]. On the other hand, we noted an extreme case where the changed parathyroid gland appeared in the soft palate of a 57-year old patient who suffered from persistent hypercalcemia even after initial parathyroidectomy. The investigation of this patient also shows that there is still a possibility that parathyroid adenomas can be missed even with previous use of advanced imaging methods if they occur in various anatomical positions [9]. Moreover, ectopic parathyroid glands can be so small that they may be invisible, or a single removed gland is one when all function abnormally [5,7]. In our opinion, the best surgical solution for the latter is a removal of three and a half of the parathyroid glands. Re-operation is also required when one of two affected glands is more active than the other. In such a situation, when the dominant parathyroid gland has been removed, the other becomes the cause of hyperfunction. A second operation is also needed when the altered parathyroid gland has not been completely removed—for example, when undergoing disintegration—and the rest of the parathyroid gland develops again causing symptoms of hyperactivity.

Most of the reasons for the failure of initial parathyroidectomy may be eliminated by the right diagnostic measures in the form of imaging possibilities [10,11]. Radiological examination methods are particularly precise and helpful to properly localize most of hyperfunctioning parathyroid glands, thus allowing for shorter surgeries, a decrease in ineffective surgeries and fewer complications [12,13]. Some studies show that in primary hyperparathyroidism the results of preoperative imaging examinations do not allow to differentiate between a uniglandular and multiglandular disease in [14,15]. Finding the parathyroid gland facilitates the intraoperative examination of parathyroid hormone concentration, as well as the use of radio-guidance [16,17,18].

Cases with multiple tumours are particularly more complicated, as a fall in the intraoperative PTH to the lower end of a normal range cannot guarantee that all tumours have been removed [19,20,21]. A reduction in intraoperative PTH by more than 50% after the procedure is a predictive factor [22,23] and may be prevented by preoperative supplementation with bisphosphonates [24,25]. Age is not a relevant contributing factor for the occurrence of complications [26,27], while anesthesia can often cause sudden elevation of PTH levels in patients suffering from hyperparathyroidism [28,29] and can worsen renal functions [30,31]. The latest works shows that minimally invasive radio-guided parathyroidectomy using a very low dose of Tc-99 MIBI, even without an intraoperative assay or a frozen section analysis, resulted in an excellent cure rate [32,33]. Based on our results, we also wanted to emphasize that PTX is a safe procedure, with only a small number of complications, the most serious of which seems to be recurrent laryngeal nerve palsy. Based on our observations, we believe that patients should be examined every three months after PTX for the first two years, and then twice a year afterwards. Some authors indicate that this should last for even more than 10 years after parathyroidectomy [34].

## 6. Conclusions

(1)Our results confirm that parathyroidectomy is the treatment of choice in patients with primary hyperparathyroidism.(2)Ectopic position of the parathyroid gland in the mediastinum is associated with an increased risk of surgical failure.(3)Most parathyroid lesions in the mediastinum can be safely removed from the cervical access.

## Figures and Tables

**Figure 1 jcm-09-03540-f001:**
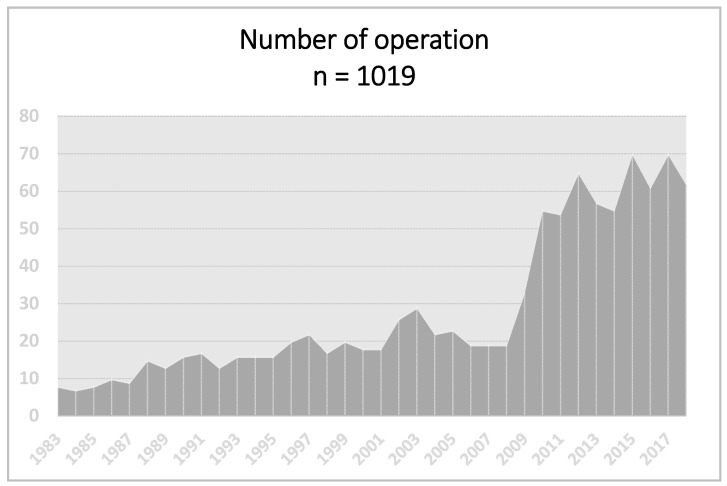
Number of operations in years 1983–2018.

**Figure 2 jcm-09-03540-f002:**
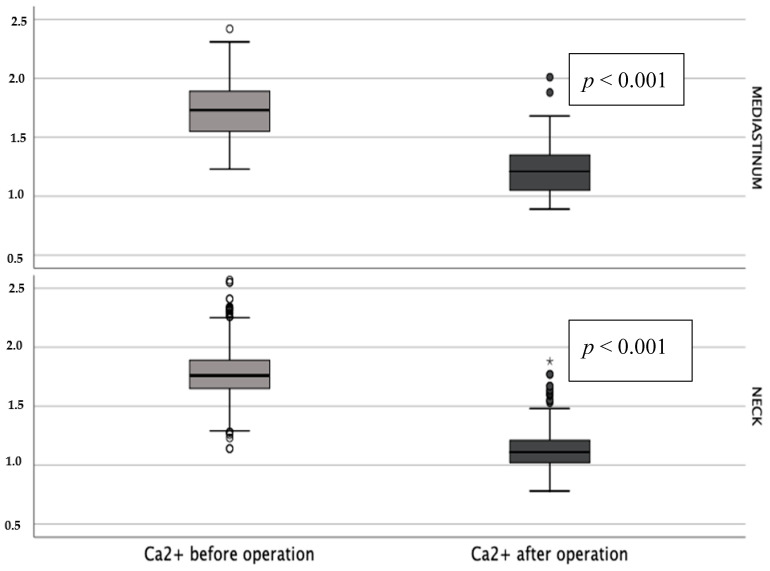
Blood serum Ca^2+^ concentration before and after operation (*—*p* < 0.001).

**Figure 3 jcm-09-03540-f003:**
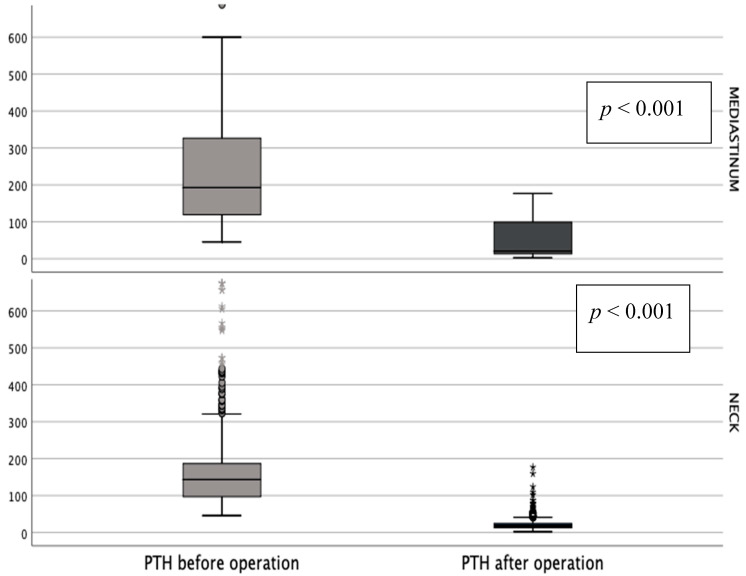
Blood serum parathormone (PTH) concentration before and after operation (*—*p* < 0.001).

**Table 1 jcm-09-03540-t001:** Symptoms of 1019 patients operated on for Primary hyperparathyroidism (pHPT).

Symptom or Sign	Whole Group of Patients (*n* = 1019)	Patients with Neck Localization (*n* = 953)	Patients with Mediastinal Localization (*n* = 66)	χ^2^	*p* Value	φ
Bone and muscle pain	797 (78.4%)	750 (78.7%)	47 (71.2%)	2.03	0.154	0.05
Arterial hypertension	493 (48.5%)	480 (50.4%)	13 (19.7%)	23.25	<0.001	0.15
Nephrolithiasis	385 (37.9%)	359 (37.7%)	26 (39.4%)	0.08	0.780	0.01
Osteoporosis	245 (24.1%)	229 (24%)	16 (24.2%)	<0.01	0.969	<0.01
Fatigue	242 (23.8%)	227 (23.8%)	15 (22.7%)	0.04	0.840	<0.01
Constipation	226 (22.2%)	211 (22.1%)	15 (22.7%)	0.01	0.912	<0.01
Mental depression	162 (15.9%)	152 (15.9%)	10 (15.2%)	0.03	0.864	<0.01
Chronic renal insufficiency	145 (14.3%)	153 (16.1%)	9 (13.6%)	0.27	0.603	0.02
Weight loss	129 (12.7%)	121 (12.7%)	8 (12.1%)	0.02	0.892	<0.01
Stomach ulcer/Duodenal ulcer	113 (11.1%)	106 (11.1%)	7 (10.6%)	0.02	0.897	<0.01
Pathological bone fracture	97 (9.5%)	91 (9.5%)	6 (9.1%)	0.02	0.902	<0.01
Acute or chronic pancreatitis	80 (7.9%)	75 (7.9%)	5 (7.6%)	<0.01	0.932	<0.01
Loss of appetite	275 (27%)	261 (27.4%)	14 (21.2%)	1.20	0.274	0.03
Dermatopathies	32 (3.1%)	23 (2.4%)	9 (13.6%)	25.56	<0.001	0.16
Without symptoms	96 (9.4%)	92 (9.7%)	4 (6.1%)	0.93	0.334	0.03

**Table 2 jcm-09-03540-t002:** Patient’s characteristics.

Factor			No. of Patients/Localization			*p* Value	ES
			All	Neck	Mediastinum		
			*n* = 1019	*n* = 953	*n* = 66		
sex	M		260	244	16	0.806	0.01
	F		759	709	50		
age (years)	M		21–79	21–79	25–79	0.240	0.01
			(55.0)	(55.0)	(55.9)		
	F		19–81	19–81	28–80	0.339	<0.01
			(55.0)	(55.4)	(56.2)		
symptoms	Yes		923	861	62	0.334	0.03
	No		96	92	4		
lesion	benign		990	928	62	0.104	0.05
	malignant		29	25	4		
no. of lesions	single		856	807	49	0.070	0.08
	multiple		163	146	17		
	2		129	114	15		
	3		26	24	2		
	4		8	8	0		
localization	left	superior	220	210	10	0.015	0.10
		inferior	351	315	36		
	right	superior	219	213	6	0.014	0.10
		inferior	436	403	33		
histopathology *n* = 1226	hyperplasia		827	772	55	0.460	0.05
	adenoma		370	344	26		
	cancer		29	25	4		
Recurrent nerve palsy			11	7	4	<0.001	0.2
transient			9	7	2		
permanent			2	0	2		
Operation failure			19 (1.7%)	8 (0.8%)	11 (16.7%)	<0.001	0.26

**Table 3 jcm-09-03540-t003:** Histopathological findings.

Number of Affected Parathyroid Glands	Histopathological Finding	Patients/Abnormalities			χ^2^	*p* Value	φ
		All	Neck	Mediastinum			
Single	Hyperplasia	492	464	28	2.44	0.486	0.05
	Adenoma	337	319	18			
	Cancer	27	24	3			
	All	856	807	49			
Two or More	2 × hyperplasia + cancer	1	1	0	0.07	0.792	<0.01
	1 × hyperplasia + cancer	1	0	1	14.45	<0.001	0.12
	2 × hyperplasia	82	73	9	2.98	0.084	0.05
	2 × adenoma	8	5	3	12.81	<0.001	0.11
	3 × hyperplasia	20	18	2	0.42	0.518	0.02
	hyperplasia + adenoma	36	34	2	0.05	0.819	<0.01
	4 × hyperplasia	5	5	0	0.35	0.560	<0.01
	2 × hyperplasia + adenoma	6	6	0	0.42	0.520	<0.01
	3 × hyperplasia + adenoma	3	3	0	0.21	0.648	<0.01
	2 × adenoma + hyperplasia	1	1	0	0.07	0.792	<0.01
sum		163	146	17	5.00	0.025	0.07

**Table 4 jcm-09-03540-t004:** Treatment failures.

Localization	Number of Patients	Treatment Failures	χ^2^	*p* Value	φ
All	1019	19 (1.9%)	21.15	<0.001	0.15
Neck	953	8 (0.8%)			
Mediastinum	66	11 (16.6%)			

**Table 5 jcm-09-03540-t005:** Persistent and recurrent hypercalcemia.

Localization	Number of Patients	Hypercalcemia		Changes Not Found	χ^2^	*p* Value	φ
		Persistent	Recurrent				
All	1019	9 (0.9%)	7 (0.7%)	3 (0.3%)	2.05	0.152	0.36
Neck	953	2 (0.2%)	4 (0.4%)	2 (0.2%)			
Mediastinum	66	7 (10.6%)	3 (4.5%)	1 (1.5%)			

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
