# Peer review of "Surgical Management of Primary Hyperparathyroidism—Clinicopathologic Study of 1019 Cases from a Single Institution"

_jcm, 2020, doi:10.3390/jcm9113540_

Round 1
Reviewer 1 Report
The text could be shortened throughout for clarity and readability.
This will generally confirm existing literature and is not novel.
Missing current reference on recurrent HPT and revision parathyroid surgery was omitted.
Stack BC Jr, Tolley NS, Bartel TB, Bilezikian JP, Bodenner D, Camacho P, Cox JPDT, Dralle H, Jackson JE, Morris JC 3rd, Orloff LA, Palazzo F, Ridge JA, Scott-Coombes D, Steward DL, Terris DJ, Thompson G, Randolph GW. AHNS Series:
Do you know your guidelines? Optimizing outcomes in reoperative parathyroid surgery: Definitive multidisciplinary joint consensus guidelines of the American Head and Neck Society and the British Association of Endocrine and Thyroid Surgeons. Head Neck. 2018 Aug;40(8):1617-1629. doi: 10.1002/hed.25023. Epub 2018 Aug 2. PMID: 30070413.
Author Response
Answer to Reviewer 1
Dear reviewer,
The whole manuscript has been shortened and modified. Now we believe, that this is more clair and readable. The missing reference has been added (position 5).
Reviewer 2 Report
The retrospective study by Gawrychowski and coworkers describes a large series of patients (n = 1019) submitted to parathyroidectomy for primary hyperparathyroidism at the same institution through three decades. As it happens with large "real life"-based retrospective studies, the study contains a large amount of potentially interesting data, likely to raise several clinically relevant questions.
Yet, the manuscript in its present form has a number of limitations and flaws (see the points below) which antagonize the full comprehension of its contents.
a) The abstract as well as the introduction need to be more straightforward and coherent. For instance, the conclusions of the abstract are explicitely focused on the issue of ectopic parathyroid (PT) glands and the limitations of localization-targeted imaging studies; nevertheless the background is apparently focused on the problem of persistent/recurrent hypercalcemia after parathyroidectomy. Furthermore, the introduction contains some paragraphs (lines 46-50) centered on the differential diagnosis between hyperplasia and adenoma, which is quite a different issue from the problem of ectopic PT glands.
b) The study period covers a long time span (namely 30 years). It is reasonable that along three decades something changed in the diagnostic and therapeutic procedure performed at the authors'institution on patients with pHPT. These changes are likely to reverberate on the clinical outcomes. The authors should take into account these considerations and the data analysis and hence the Material and Methods section should be accordingly modified.
c) In the Results section, symptoms and signs of pHPT as well as data concerning the results of imaging studies for the localization of abnormal PT glands should be listed before (and not after) reporting the histological findings.
d) The discussion as well as the Conclusions need to be shortened and kept adherent to the findings of the study.
e) The English Language needs a deep revision. For clarity purpose, many sentences need also to be modified. Here are some "non-exhaustive" examples:
Line 37 "abnormally changed parathyroid glands"; I would suggest as follows: "abnormal parathyroid glands"
Line 38 "within the organism": redundant, it can be profitably omitted.
Line 39 risk factor for developing etc.
Line 43 "Most parathyroid changes involve a single gland in a normal neck localization"; I would suggest as follows: "Most cases involved a single abnormal parathyroid gland located in an usual (eutopic) neck site".
Line 45 or soft instead "of soft"
Line 55 Focused parathyroidectomy is the procedure of choice
LIne 68 "All patients had undergone an evaluation before referral to our department, but the responsibility to confirm the diagnosis pHPT remained with us". This sentence sounds quite obscure. I would suggest to simply mention that "1019 patients diagnosed with primary hyperparathyroidism underwent parathyroid surgery at our Institution in the period etc etc"
Line 87 "Women were affected almost three times more often than men". This sentence is incorrect: this is not epidemiological study aimed to define the prevalence of hyperparathyroidsm in the general population. Rather, the study population featured, as exptected, a majority of female subject. In addition, Any comment to this line of evidence should be included in the discussion.
Line 88 "Displaced parathyroid glands were found sporadically in the mediastinum (6.4%) and in the vast majority of cases were localized in the neck area (93.6%)" This sentence should be rephrased in order to make it more fluent and understandable, e.g. "The vast majority (93.6%) of parathyroid lesions were localized in the neck, whereas a minority (6.4%) was found in the mediastinum"
Line 90 Suggested version: "Histological findings demonstrated benign lesions in 990 patients while malignancy was reported in the remaining 29 cases".
lines 92-96 I would suggest to change the text as follows: "As a whole, 1226 parathyroid glands were excised and examined in the 1019 patients of our series (table 1). Among the patients who had multiple PT lesions, two affected glands were the most frequent event, while four affected glands were a sporadic finding"
Author Response
Answer to Reviewer 2
Dear reviewer,
a) Both the abstract and the introduction have been completely revised. We changed the conclusion, and now we believe that they are more straightforward and coherent.
b) The material and methods section has been modified. We divided this period of time into parts, and added a figure that shows number of patients in every year from 1983 to 2018.
c) In the Results section, symptoms and signs of pHPT as well as data concerning the results of imaging studies for the localization of abnormal PT glands should be listed before (and not after) reporting the histological findings. – The Results section has been modified.
d) The discussion as well as the Conclusions need to be shortened and kept adherent to the findings of the study. – Both section has been shortened and modified.
e)
Line 37 "abnormally changed parathyroid glands"; I would suggest as follows: "abnormal parathyroid glands" – this sentence has been changed (Line 37)
Line 38 "within the organism": redundant, it can be profitably omitted. - this sentence has been changed
Line 39 risk factor for developing etc. – Line 33
Line 43 "Most parathyroid changes involve a single gland in a normal neck localization"; I would suggest as follows: "Most cases involved a single abnormal parathyroid gland located in an usual (eutopic) neck site". – Line 36
Line 45 or soft instead "of soft" – Line 39
Line 55 Focused parathyroidectomy is the procedure of choice – Line 44
LIne 68 "All patients had undergone an evaluation before referral to our department, but the responsibility to confirm the diagnosis pHPT remained with us". This sentence sounds quite obscure. I would suggest to simply mention that "1019 patients diagnosed with primary hyperparathyroidism underwent parathyroid surgery at our Institution in the period etc etc" – The Material and Methods section has been modified.
Line 87 "Women were affected almost three times more often than men". This sentence is incorrect: this is not epidemiological study aimed to define the prevalence of hyperparathyroidsm in the general population. Rather, the study population featured, as exptected, a majority of female subject. In addition, Any comment to this line of evidence should be included in the discussion.
This sentence has been deleted.
Line 88 "Displaced parathyroid glands were found sporadically in the mediastinum (6.4%) and in the vast majority of cases were localized in the neck area (93.6%)" This sentence should be rephrased in order to make it more fluent and understandable, e.g. "The vast majority (93.6%) of parathyroid lesions were localized in the neck, whereas a minority (6.4%) was found in the mediastinum"
Line 89
Line 90 Suggested version: "Histological findings demonstrated benign lesions in 990 patients while malignancy was reported in the remaining 29 cases". – Line 91
lines 92-96 I would suggest to change the text as follows: "As a whole, 1226 parathyroid glands were excised and examined in the 1019 patients of our series (table 1). Among the patients who had multiple PT lesions, two affected glands were the most frequent event, while four affected glands were a sporadic finding" – Line 92-94
Round 2
Reviewer 2 Report
All the most relevant points of criticism have been addressed by the authors. Besides some suggested changes to the text have been included. In its present form the paper sounds more clear and fluent.
Just some minor observations:
a. Table 1, 2nd and 3rd columns: "patients neck localization" and " patients mediastinal localization" should be made more clear. I would suggest to insert the preposition "with", i.e. "patients with neck localization" and "patients with mediastinal localization", but other options may be effective.
b. The following sentence (lines 89-90) "The vast majority (93.6%) of parathyroid lesions were localized in the neck (predominantly within the left and right superior glands), whereas a minority (6.4%) was found in the mediastinum." should profitably be moved before the previous paragraph, i.e. the one starting at line 85 ("We observed a significant difference etc.")
c. Conclusion, line 208 2)....increased risk of surgery. I guess "increased risk of surgical failure" should be the right version.
d. I would add just a brief sentence in the Materials and Methods section, where the localization studies are listed (lines 58-62), in order to explain which criteria had been followed to consider the imaging pre-surgical work up as "positive". For instance, were two positive imaging studies always required? Was instead just a single imaging study consider sufficient, when clearly positive? One could argue that a possible cause of surgical failure may be explained by an erroneous or ambiguous imaging work-up.
Author Response
Dear reviewer,
Thank you for your suggestions, all of them has been changed.
Kind regards
Authors
